

# Open surgery versus retroperitoneal laparoscopic nephrectomy for renal tuberculosis: a retrospective study of 120 patients

Su Zhang[*], You Luo[*], Cheng Wang, Hu Xiong, Sheng-Jun Fu and Li Yang

Department of Urology, Lanzhou University Second Hospital, Lan Zhou, China
[*] These authors contributed equally to this work.

## ABSTRACT

**Background**. Laparoscopic renal surgery has been widely used in the treatment of renal diseases. However, there is still little research about its application in addressing renal tuberculosis. The purpose of this study is to retrospectively investigate the surgical results of laparoscopic and open surgery for nonfunctional tuberculous kidneys.

**Methods**. Between May 2011 and June 2016, 120 nephrectomies were performed in patients with a nonfunctional tuberculous kidney. Of these, 69 patients underwent retroperitoneal laparoscopic nephrectomy, and 51 patients underwent open nephrectomy. Data about the patients' characteristics and surgical outcomes were collected from their electronic medical records. Outcomes were compared between these two groups.

**Results**. Our results showed that a number of renal tuberculosis patients presented no significant symptoms during their disease. Lower urinary tract symptoms (LUTS) were the most common at a rate of 73/120, followed by flank pain or accidently discovery (66/120), urine abnormality (30/120) and fever (27/120). Patients who underwent open surgery were similar to laparoscopic patients with regard to sex, BMI, location, previous tuberculous history, grade, anemia, adhesion, hypertension, diabetes and preoperative serum creatinine level, but were generally older than laparoscopic patients. There were no significant differences between open and laparoscopic surgery in estimated blood loss, transfusion, postoperative hospital days and perioperative complication rate. However, the median operation time of laparoscopic operation was much longer than open surgery (180 [150–225] vs 135 [120–165] minutes, $P < 0.01$). Seven of the 69 laparoscopic operations were converted to open surgery because of severe adhesions.

**Conclusion**. Laparoscopic nephrectomy is as an effective treatment as open surgery for a nonfunctional tuberculous kidney, although it requires more time during the surgical procedure. No significant differences in other surgical outcomes were observed.

Corresponding author
Li Yang, professoryangli@163.com

## INTRODUCTION

In recent years, the resurgence of tuberculosis has become a subject of global interest, particularly for developing countries, due to drug resistance of the *Mycobacterium tuberculosis* and population mobility (*World Health Organization, 2015*). Renal tuberculosis is a kidney-destructive disease and comprises one of the most common types of extrapulmonary tuberculosis (*Fader et al., 2010*). In many patients, the disease has already progressed to the end stages before they consult their doctors. The standard treatment for a nonfunctional tuberculous kidney is a nephrectomy combined with anti-tuberculosis chemotherapy. The laparoscopic technique was considered contraindicative for resecting a kidney with tuberculosis because the disease caused severe adhesion and perinephritis. However, with the development of new techniques and increased experience, the laparoscopic operation is no longer contraindicated for experienced laparoscopic surgeons (*Kim et al., 2000*; *Lee et al., 2002*). In this paper, we aim to summarize recent developments in nephrectomies and compare the outcomes of laparoscopic operations with open surgery.

## MATERIAL AND METHODS

### Patient population

From May 2011 to June 2016, surgical records from patients treated by nephrectomy with unilateral nonfunctioning kidney secondary to renal tuberculosis were collected from the Department of Urology in Lanzhou University Second Hospital in this case-control study, after the approval of the Ethics Committee of Lanzhou University Second Hospital (Number: 2016A-076). Preoperative laboratory (e.g., urine smear for acid-fast bacilli, urine polymerase chain reaction for acid-fast bacilli) and imaging examinations (e.g., renal ultrasonography, computerized tomography, intravenous pyelogram), were conducted in patients suspected of renal tuberculosis, who were also conformed finally by postoperative pathological diagnosis. The patients who had abdomen operation history or underwent other surgeries simultaneously were excluded from the study.

In total, 120 consecutive patients including 51 patients underwent open nephrectomies and 69 patients underwent laparoscopic nephrectomies were enrolled in this study. Each tuberculous patient received anti-tuberculosis chemotherapy two weeks prior to surgery and was prescribed six to nine months of postoperative chemotherapy (5 mg/kg isoniazid orally once daily, 10 mg/kg rifampicin orally once daily, 15 mg/kg ethambutol orally once daily, with or without 10 mg/kg pyrazinamide orally twice daily). All of the surgeries were performed using the retroperitoneal approach by three senior surgeons who had performed at least 50 nephrectomies with each of the two surgical approaches. The kidneys were evaluated before surgery and confirmed to be nonfunctional by intravenous urography or radionuclide renogram. The surgical approaches (laparoscopic vs open) were selected mainly depending on patient and surgeon preference. The decision to convert to open surgery was made by the surgeons during the procedure. The basic characteristics of patients included in the study included their primary symptoms including their chief complaint, age, sex, body mass index (BMI), resected kidney location (right or left), previous history of tuberculosis, adhesion of perirenal tissue, preoperative serum creatinine level,

anemia, hypertension, diabetes and their American Society of Anesthesiologists (ASA) grade. Electronic medical records and the designated entries for "Chief complaint" and "Present illness" were used to collect data on biographical information and medical history. The patient's ASA grade was obtained from the anesthesiology records, and their adhesion status was gathered from descriptions by the operating surgeon. Additional information was gathered regarding the following surgical-relevant outcomes: operation time, estimated blood loss, administration of transfusion, postoperation hospital days, perioperative complications, and conversion to open surgery.

## Outcomes and statistics

Perioperative complications were defined as any of the following symptoms or conditions: fever, abdominal distention or ileus, abscess or infection of the incision area, and seroma volume of drainage with more than 200 ml and classified according to the Clavien–Dindo grading system (*Dindo, Demartines & Clavien, 2004*) Normality was measured using the Shapiro–Wilk test. Non-normal parameters were presented as median ranges (interquartile range: IQR), and the Mann–Whitney test was used to test the difference. Normal outcomes were presented as the mean $\pm$ standard deviation (SD) and independent $t$-test samples were used for comparison. Chi-square test was used for counting data, and MedCalc Statistical Software version 15.5 (MedCalc Software bvba, Ostend, Belgium) was used to perform statistical evaluations. A two-sided $P < 0.05$ was considered statistically significant.

## RESULTS

The basic characteristics of included patients are shown in Table 1. Open nephrectomy patients with a mean age of 44.57 years old were greater than laparoscopic patients with a mean age of 39.06 years old ($P = 0.02$). Only four were over 65 years old, and the youngest was 17 years old. The age of entire cohort ranged from 17 to 67 years old. Previous tuberculosis history was present in 20 patients. There were 59 patients with moderate to severe adhesion and 26 patients with anemia. No statistical significance was observed between open nephrectomy and laparoscopic nephrectomy by sex, BMI, surgical location, ASA grade, moderate to severe adhesion rate, anemia, hypertension, diabetes, or preoperative serum creatinine level. Additionally, data about the symptoms patients indicated as their chief complaint was collected and classified into four categories: lower urinary tract symptoms (LUTS), flank pain or accidently discovery, urine abnormality (hematuria or cloudy urine) and fever. The rate of each symptom was demonstrated as LUTS (73/120), flank pain or accidently discovery (66/120), urine abnormality (30/120) and fever (27/120). It is noteworthy that 23 of the 120 patients only complaint of flank pain or fatigue and seven cases were accidently discovered. Renal tuberculosis is often asymptomatic and is difficult to diagnose.

Table 2 lists the surgical outcomes of the two treatment options. Seven laparoscopic cases were converted to open nephrectomy. The median operation time was 135 min (IQR: 12,0165 min) for open surgery and 180 min (IQR: 150–225 min) for laparoscopic nephrectomy ($P < 0.01$). All postoperative complications were grade 1 according to Clavien–Dindo classification. There was no significant difference between open

**Table 1  Characteristics and comparative results of open versus laparoscopic surgery for renal tuberculosis.**

| | Total | Surgery | | P value |
| --- | --- | --- | --- | --- |
| | | Open nephrectomy | Laparoscopic nephrectomy | |
| No. of pts | 120 | 51 | 69 | – |
| Age (years) | 41.40 ± 12.55 | 44.57 ± 13.0 | 39.06 ± 11.74 | 0.02[***] |
| Sex (female/male) | 55/65 | 27/24 | 28/41 | 0.18[*] |
| BMI | 22.6 ± 3.35 | 22.39 ± 2.95 | 22.76 ± 3.63 | 0.55[***] |
| Location (left/right) | 63/57 | 27/24 | 36/33 | 0.93[*] |
| Previous tuberculous history (yes/no) | 20/100 | 9/42 | 11/58 | 0.80[*] |
| ASA grade | | | | |
| I | 8 | 5 | 3 | |
| II | 106 | 42 | 64 | 0.21[*] |
| III | 6 | 4 | 2 | |
| Moderate-to-severe adhesion (yes/no) | 59/61 | 28/23 | 31/38 | 0.28[*] |
| Anemia (yes/no) | 26/94 | 12/39 | 14/55 | 0.67[*] |
| Hypertension (yes/no) | 11/109 | 3/48 | 8/61 | 0.35[**] |
| Diabetes | 5/115 | 2/49 | 3/66 | >0.99[**] |
| Preoperative Scr | 83.65 (70–95.55) | 82 (67–91) | 85 (73–100.5) | 0.12[****] |

Notes.

Annotation: the interval in the parenthesis is interquartile range and the number before parenthesis is the median value; Scr- serum creatinine;

[*] Chi-square test.

[**] Fisher's exact test.

[***] Independent samples $t$-test.

[****] Mann–Whitney test.

**Table 2  Surgical outcomes of the different operation methods.**

| | Total | Open nephrectomy | Laparoscopic nephrectomy | P value |
| --- | --- | --- | --- | --- |
| No. of pts | 120 | 51 | 69 | – |
| Operation time (minutes) | 160 (132–198.75) | 135 (120–165) | 180 (150–225) | <0.01[***] |
| Estimated blood loss (ml) | 100 (50–200) | 100 (50–150) | 100 (50–200) | 0.69[***] |
| Transfusion (yes/no) | 9/111 | 1/50 | 8/61 | 0.08[**] |
| Postop hospital days | 8 (7–10.75) | 9 (7–11) | 8 (7–10) | 0.11[***] |
| Perioperative complication rate | 37/120 | 14/51 | 23/69 | 0.49[*] |
| Conversion to open surgery | – | – | 7/69 | – |

Notes.

[*] Chi-square test

[**] Fisher's exact test

[***] Mann–Whitney test

nephrectomy and laparoscopic nephrectomy in estimated blood loss, transfusion rate, postoperative hospital days and perioperative complication rates (Table 2).

## DISCUSSION

The 2015 global tuberculosis report showed that there were an estimated 9.6 million new TB cases and China accounted for 10% of the world's TB cases ranking second in the world (*World Health Organization, 2015*). According to the National Health and Family Planning Commission of PRC, there were 95924 new TB cases in August (*PRC, 2016*) and

1918 new cases in Gansu province citing the Health and Family Planning Commission of Gansu (*Gansu, 2016*) As the data show, the people in Gansu have a high incidence of tuberculosis, and furthermore seek medical advice with relative delay. The manifestation of tuberculosis is increasingly atypical. Unfortunately, the prevalence of drug-resistant strains of the tuberculosis bacterium is increasing (*Lee et al., 2015*). Latent tuberculosis presents great challenges in terms of diagnosis and treatment (*Getahun et al., 2015*). Renal tuberculosis is a severe organ-destroying disease. The majority of symptomatic outpatients has progressed to the end stages and has a nonfunctional kidney. According to the 2005 EU guideline, the drug treatment is the first line therapy for renal tuberculosis (*Cek et al., 2005*). However, patients in Gansu usually present end stage and the surgical excision of non-functioning kidneys or highly destructive lesion is unavoidable.

As our results describe, 23 of 120 patients only presented with flank pain or fatigue, symptoms they had previously ignored, and seven patients were discovered accidently. Huang et al. surveyed 239 cases of renal tuberculosis. In their study, 94 of the 239 renal tuberculosis cases were atypical. The main symptoms experienced by the typical group were lower urinary tract symptoms, flank pain, hematuria and fever, etc. In addition, flank pain was regarded as the major complaint in the atypical group (*Huang, Li & Jin, 2013*). Wang et al. also confirmed these findings (*Wang et al., 2016*). A previous history of tuberculosis was observed in twenty patients in our cohort. Huang et al. reported that nearly 50% (125/239) of renal tuberculosis patients had a tuberculosis history or extrarenal findings in radiological images (*Huang, Li & Jin, 2013*). Therefore, we can conclude that tuberculosis history is a good clue for the diagnosis of renal tuberculosis (*Cek et al., 2005*).

The use of the laparoscopic technique has become very popular among surgeons for dealing with renal diseases. However, renal tuberculosis challenges this application because of its severe adhesion and perinephritis. Both of these obstacles could contribute to the need to convert to open surgery during the laparoscopic procedure. All surgery was performed by retroperitoneal laparoscopic approach Compared with transperitoneal laparoscopic nephrectomy, retroperitoneal laparoscopic nephrectomy has several advantages, particularly for patients with normal anatomy (*Fan et al., 2013*; *Garg et al., 2014*; *Zhang et al., 2013*). Although one study reported 10 laparoscopic nephrectomies using the transperitoneal approach and 21 using the retroperitoneal approach, the comparison between the two approaches was not conducted in the results (*Lee et al., 2002*) Hence, the advantages and disadvantages between the two laparoscopic approaches could not be identified from previous researches. In 2000, Hemal et al. compared nine retroperitoneoscopic nephrectomies with nine open nephrectomies for a nonfunctional tuberculous, kidney and their work showed no significant differences between operative times, blood loss and complications. Retroperitoneal nephrectomy is associated with significantly shorter hospital days and recovery time, and less analgesic is requirement than in open surgery (*Hemal, Gupta & Kumar, 2000*; *Hemal et al., 2007*; *Zhang et al., 2005b*). Unfortunately, our study could not calculate the difference in the levels of analgesics used for postoperative pain therapy because patientcontrolled intravenous analgesia was used in some postoperative patients. In addition, our results show a longer operation time in laparoscopic group (median 180 min) than open group (median 135 min) that is

consistent with the result of Hemal's study to a certain extent (*Hemal, Gupta & Kumar, 2000*). However, compared with existing reports in which surgical time of laparoscopic approach ranged from mean 92 min to mean 244 min (*Lee et al., 2002*; *Tian et al., 2015*; *Zhang et al., 2005b*), the median time of our operations was 180 min and this difference may be due to the selection of patients or the surgical experience.

In 1998, among five patients with renal tuberculosis treated by laparoscopic nephrectomy, four patients (80%) suffered the conversion to open surgery because of difficult dissection of the severe adhesions (*Rassweiler et al., 1998*). Then *Zhang et al. (2005a)* reported 12 successful retroperitoneoscopic nephrectomies without conversion to open operation for an infective nonfunctioning kidney with dense perinephric adhesions. Additionally, they also confirmed the operability and advantages of retroperitoneal laparoscopic nephrectomy for renal tuberculosis by comparing 22 successful laparoscopic with 22 open nephrectomies (*Zhang et al., 2005b*) Recently *Tian et al. (2015)* reported 51 consecutive patients with tuberculous nonfunctioning kidney treated by laparoscopic nephrectomy and there was only one case required conversion to open surgery due to non-progression of dissection. The present research presented nearly 10 percent of conversion to open surgery and the good news is all these surgical conversion resulted from uncompleted dissection of the severe adhesions without any vascular injury or visceral injury. Thus, perinephric adhesions should not be considered an absolute contraindication to laparoscopic nephrectomy and the comprehensive preoperative evaluation, the delicate surgical technique and the accumulation of surgical experiences are very important to complete laparoscopic nephrectomy for tuberculous nonfunctioning kidneys successfully. There were some discrepancies within our study, and the results could thus be attributed to sample size and a more recent time period. We have noted that this study was a retrospective. Selection bias was unavoidable. Randomized controlled trials with a larger sample size are needed to confirm these results.

## CONCLUSION

Laparoscopic nephrectomy is as an effective treatment as open nephrectomy for a nonfunctional tuberculous kidney. Although the laparoscopic procedure takes longer than the open surgery, no significant differences in other surgical outcomes were observed.

### Funding

The authors received no funding for this work.

### Competing Interests

The authors declare there are no competing interests.

### Author Contributions

- Su Zhang conceived and designed the experiments, performed the experiments, analyzed the data, contributed reagents/materials/analysis tools, wrote the paper.

- You Luo conceived and designed the experiments, performed the experiments, analyzed the data, wrote the paper.
- Cheng Wang prepared figures and/or tables.
- Hu Xiong performed the experiments, prepared figures and/or tables.
- Sheng-Jun Fu and Li Yang reviewed drafts of the paper.

## Human Ethics

The following information was supplied relating to ethical approvals (i.e., approving body and any reference numbers):

Ethics Committee of Lanzhou University Second Hospital
Number:2016A-076.

## Data Availability

The raw data has been supplied as a Data S1.

## Supplemental Information

Supplemental information for this article can be found online at http://dx.doi.org/10.7717/peerj.2708#supplemental-information.

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
