# Peer review of "Open surgery versus retroperitoneal laparoscopic nephrectomy for renal tuberculosis: a retrospective study of 120 patients"

_PeerJ, doi:10.7717/peerj.2708_

## Round 0.1 · original submission · Major Revisions

Nicely written paper, but as can be read in the reviewers comments needs further work.
Especially on patient selection and how choice of treatment was determined. Also on outcome essential information is lacking.

Conclusions need to be discussed in more detail and compared with previously published work

·

Basic reporting

No comments

Experimental design

No comments

Validity of the findings

No comments

Additional comments

The current study is an interesting retrospective case-control study. The reviewer has no major comments on the design of the study. However, I do have several suggestions to improve the manuscript:
- It should be stated in the material and methods section that this is a case-control study
- The selection process of cases of nephrectomy for TBC within the hospital remains unclear and should be further explained. How many nephrectomies were performed during the 5 year period in general (other indications such as RCC). Were cases selected based on the nephrectomy specimens (pathological signs of TBC; i.e. mycobacteria/granuloma’s), or was TBC already diagnosed preoperatively, for example in urine culture in the asymptomatical patients. Nowhere in the manuscript it was even stated that pathological examination of the resection specimens was performed, not to mention retrospective chart review of the pathology report.
- As there was no major difference in the rate of adhesions between men who received laparoscopic vs open surgery (45% vs 55%, p = 0.28) it should be further elucidated why men received the specific surgical approach. How many surgeons operated on the 120 men? Did some surgeons operate laparoscopically and others open? Or was there an institutional evolution from open nephrectomy towards the laparoscopic approach within the 5 year time period? Were the open nephrectomies performed retroperitoneal (lumbotomy) or sometimes transperitoneal?
- 120 cases of nephrectomy for TBC within 5 years seems a lot for clinicians who practice in areas where (extrapulmonary/renal) TBC is very rare. Therefore, it might be helpful to add some background information in the discussion section on the incidence of extrapulmonary/renal TBC in China in general and in the Gansu Provence in particular.
- The conclusion of the authors is that the laparoscopic approach is as effective as the open approach: no significant difference in blood loss, transfusion rate and postoperative hospital days was found. In line with the results of larger studies comparing the open and laparoscopic approach for other indications (RCC, UPJ stenosis), one might expect to see an advantage of the laparoscopic approach. Considering the longer operating time with the laparoscopic approach and the 10% conversion rate, are these results not more or less in favor of the open approach (not taking into account postoperative pain)? Please discuss.

Reviewer 2 ·

Basic reporting

- The manuscript is nicely written and does not contain any obvious grammatical errors
- The introduction an background looks sufficient.
- No comparrison with already available data (e.g. Xian et al, 2015).

Experimental design

- The research question was well described. Aswell as the gap of knwoledge.
- This study unfortuntaley is a retrospective study, for this disease I think this is the best option. Pro-spective randomized controleled studies will never meet the inclusion numbers...
- The authors do not describe who of how many surgeons performed the surgery. What was the experience of the surgeon. Did one surgeon perform both procedures? This should be clarified.
- How was the function of a kindey determined? Please describe this.
- What kind of post-operative chemotherapy was given? The same in all patients?
- Describing what operating time is, is not necessary, is obvious. Also applys for length of hospital stay.
- The adverse evemts/complications should be addressed as Clavien-Dindo to make it comparably with other studies.

Validity of the findings

- The data seems robust and statistically sound.
- Conclusion drawn are correct and seem to correctly underline the findings of this paper
- The discussion should be expanded. As described above, the authors need to compare their work with others more properly. Xian et al is missing.

Additional comments

I doubt the clinical relevance of this article. Renal tuberculosis is a disease that is not common.

---

## Round 0.2 · accepted · Accept

All reviewer comments were adequately addressed.